# Energy Channelization Analysis of Rough Tools Developed by RM-MT-EDM Process during ECSM of Glass Substrates

**DOI:** 10.3390/ma15165598

**Published:** 2022-08-15

**Authors:** Tarlochan Singh, Akshay Dvivedi, Sarabjeet Singh Sidhu, Evgeny Sergeevich Shlykov, Karim Ravilevich Muratov, Timur Rizovich Ablyaz

**Affiliations:** 1Department of Product and Industrial Design, Lovely Professional University, Phagwara 144001, Punjab, India; 2Mechanical and Industrial Engineering Department, Indian Institute of Technology Roorkee, Roorkee 247667, Uttarakhand, India; 3Mechanical Engineering Department, Sardar Beant Singh State University, Gurdaspur 143521, Punjab, India; 4Mechanical Engineering Faculty, Perm National Research Polytechnic University, 614000 Perm, Russia

**Keywords:** ECSM, EDM, tool electrode, surface roughness, energy

## Abstract

In the present work, the effect of tool surface roughness on energy channelization behavior was analyzed during the fabrication of micro holes by an electrochemical spark machining (ECSM) process. In this study, rough tools were fabricated by a rotary mode multi tip electric discharge machining (RM-MT-EDM) process. The electrical characterization was also carried out to investigate the electric field intensity over the surface of tool electrode, and it was found that the use of rough tools improves the electric field intensity by 265.54% in comparison to the smooth tool electrodes. The use of rough tools in the ECSM process forms thin and stable gas film over the tool electrode, and as a result the rough tools produced high frequency spark discharges. Energy channelization index and specific energy were considered as response characteristics. The use of rough tools improves energy channelization index by 248.40%, and the specific energy is reduced by 143.263%. The material removal mechanisms for both of the processes (RM-MT-EDM and ECSM process) have also been presented through illustrations.

## 1. Introduction

Electrochemical spark machining (ECSM) is an emerging non-traditional hybrid micro machining method which can be utilized for the subtractive processing of all materials, regardless of their thermal and electrical conductivities, hardness, and reflectivity [1,2,3]. ECSM integrates the characteristics of two non-traditional machining processes (i.e., electric spark machining (ESM) and electrochemical machining (ECM)). The basic configuration of ECSM comprises two electrodes (tool electrode (cathode) and counter/auxiliary electrode (anode)), an electrolyte chamber, and a DC power supply with straight polarity [4,5]. The DC power supply connects both the electrodes and forms the electrochemical cell circuit (ECC) across the entire components of the ECSM system [6]. The application of an electric potential across ECC results in hydrolysis of electrolyte. Subsequently, it evolves oxygen and hydrogen gas bubbles from auxiliary and tool electrodes, respectively [7,8]. The coalescence of hydrogen bubbles completely covers the surface of the tool electrode with an insulating gas film. Further, the insulating gas film breaks due to the existence of high electric potential and, consequently, initiates the spark discharges from the tool electrode. The thermal energy liberated by the spark discharges removes the undesired material from the work surface through melting and vaporization phenomena [9,10]. For effective energy utilization, the worked material should be placed as close as possible to the tool electrode. T Singh et al. recommended this working gap be zero to effectively channelize the energy between tool electrode and work material [11]. Additionally, the geometric dimensions and the form of the tool electrode play a key role in controlling the spark activities and subsequent supply of electrochemical spark (ECS) energy to work material [11]. In case of smooth tools, the spark discharge initiates from the bottom edges of the tool electrode owing to the high electric field intensity. Thereby, in the case of smooth tools, the edges of tool electrode channelize the applied energy towards the center of the tool electrode [7]. However, in such circumstances the area of work material beneath the central portion of the tool electrode remained un-machined. Herein, most of the applied energy is consumed to increase the diametric (D) dimensions of the hole instead of the hole depth [9]. Therefore, the use of smooth tools for the machining of holes with a high L/D ratio seems inappropriate. In order to overcome the above-mentioned problems, the tool electrodes in various shapes and configurations (such as a spherical tip tool electrode, abrasive coated tool electrode and tool electrode with flat side walls) have been utilized in the recent investigations [5]. The basic objective of such tool configurations was to enhance the electrolyte supply at machining zone, and to augment the uniform distribution of electric field intensity over the surface of tool electrode. However, the issues related to the fabrication of these tools limit their further usage in ECSM process.

According to Singh and Dvivedi, the tool electrode with micro-cavities assists in the growth of stable and thin gas bubbles at the bottom surface of the tool electrode [11]. The use of these tool electrodes also improves the electric field intensity due to the presence of multiple sharp edges [11]. In another investigation, Arab et al. examined the effect of rough tools during the fabrication of micro holes with ECDM process. They employed a W-EDM process to generate the rough tool electrodes. However, this process was inadequate to maintain the cylindrical shape of the tool electrodes. Moreover, Arab et al. also performed a post processing method by ECM process to minimize the heat effected areas from tool surface [12]. Thus, the direct use of W-EDM process to produce the rough tools is not recommended for the ECSM applications. Apart from W-EDM process, several researchers utilized other non-conventional machining processes to generate the rough tools such as a laser beam machining process [13], wire-electric discharge grinding [14] and an electro-chemical machining [ECM] process [15]. Nevertheless, these processes found limited applications for generating rough surfaces over the cylindrical shaped micro tool electrodes. Wire electric discharge grinding consumes lot of time to produce the rough surfaces. However, in comparison to these processes, the die sinking EDM process has been utilized in several applications (such as to develop die and molds) where a controlled and precise surface roughness is required [16,17]. However, the controlled impact of discharge energy represents a big barrier for the utilization of EDM process to fabricate micro tools. In order to control the discharge impact in the EDM process, several modifications have been provided in previous years in terms of tool configurations by employing composite material-based tool electrodes and by using different dielectric mediums based on biodegradable contents and nano-powder mixed dielectrics [18,19,20]. Nevertheless, these modifications have been reported only for the generation of rough surfaces on flat work material. Thereby, there is still need to develop new variation of EDM process to liberate the uniform and controlled energy over the tool surface in order to produce the rough surfaces.

As per the concern of the analysis of rough tools in the ECSM process, the detailed investigation with respect to surface roughness values has not yet been reported. In previous investigations, Singh et al. and Arab et al. suggested using rough tools over smooth tools, since rough tool electrodes offer better machining performance in terms of energy penetration [11,12]. They supported these results by providing the theory of the generation of small cavities at the bottom surface of tool electrode, which further helps to generate thin and stable gas film at the bottom surface. Whereas the work of Yang et al. contradicts these outcomes, they reported that with an increase in tool roughness, the gas film thickness also increases due to the accumulation of bubbles for long durations over the tool surface. As a result, the use of rough tools deteriorates the surface quality [21]. So, due to the lack of experimental investigations over the roughness scale, it becomes difficult to approximate the exact impact of rough tools in the ECSM process. Thereby, there is still a need to investigate the performance of ECSM process with tool electrodes having different values of surface roughness.

Thus, in the present study, a new method is proposed to generate rough surfaces over cylindrical tools, and experimental investigations were carried out to obtain the different roughness values. Further, these rough tools having different surface topographies (surface roughness) were used to machine micro holes in glass by ECSM process. Moreover, the energy channelization index and specific energy were considered to measure the performance of ECSM process. The methods and equipment used to conduct the investigation are described in subsequent sections.

## 2. Materials and Methods

In the present investigation, a laboratory scale ECSM facility was employed to conduct the experimental work, and the pictorial view of the same is depicted in Figure 1.

The ECSM facility comprises various sub-systems such as an electrolytic chamber, DC pulsed power supply, tool electrode, and auxiliary electrode. A cylindrical shaped stainless steel (SS-304) tool electrode having a diameter of 400 μm was used for machining. During experimentation, borosilicate glass was used as a work material, and it was positioned beneath the tool electrode at zero working gap. In order to maintain the zero working gap between work material surface and tool electrode bottom surface, and to feed the work material after material removal, a pressurized feeding system was used in this study, and the schematic view of the pressurized feeding system is shown in Figure 2. Graphite was used as an auxiliary electrode and it was placed beneath the tool electrode at a distance of 20 mm from the tool tip. In this study, sodium hydroxide (NaOH) was placed inside the electrolyte chamber to provide the conductive path across two electrodes. A pulsed DC power supply was used to connect both the electrodes (Figure 1). In the ECSM process, the voltage signals play a key to describe the behavior of spark characteristics. Thus, in this study, a digital storage oscilloscope (Agilent DSOX3034A, Santa Clara, CA, USA) was used to record the voltage signals. In these signals, the term Vmax is used to indicate the intensity of sparks.

A wire brush based novel rotary mode multi tool electric discharge machining (RM-MT-EDM) process was used to fabricate the rough surface over the cylindrical tool electrodes. The schematic and pictorial view of the RM-MT-EDM set up for tool roughening is shown in Figure 3. In this system, the copper wire brush serves as a cathode. The rotary SS-304 smooth tool electrode was connected to the positive terminal of DC pulsed power supply. The kerosene oil was used as a dielectric medium. An in-house developed rotary system was used to rotate the tool electrode, and further this system was installed over the EDM column (Electronica (EMS 5030) die sinking, Fairlie Place, Kolkata, West Bengal, India) to move the tool electrode in up and downward directions. In order to capture the surface profiles and to measure their respective roughness value in terms of Ra value, an Optical Profiler (WYKO NT1100, Tucson, AZ 85706, USA) was used in this study. The roughness was measured at five different locations which includes the side as well as the bottom surfaces of the tool electrode, and finally the average value was considered as the final response. The roughness was measured from the cross-section view of the surface profile. The measurement length was taken as 250 μm. Three reading were taken from the tool side surfaces at different locations, and these locations were selected by rotating the tool electrode at approximate 120 degree. The final two readings were taken from the bottom surface of the tool, and the location to select these reading were based on the equally divided area at the bottom face of tool.

As mentioned previously, during the ECSM process, the distribution of electric field intensity over the surface of the tool electrode plays a key role in controlling the machining performance. Thereby, during ECSM, it is necessary to characterize the distribution of electric field intensity with respect to the different topo graphs of tool electrodes generated by the RM-MT-EDM process. Thus, to characterize the electric field intensity over the tool electrode surface, COMSOL V.5.2 multi physics simulation software was used in this study. During simulation, an electrostatic model in static mode was selected. A 2D geometric model was constructed to characterize electric field intensity effect. A tool electrode was selected of stainless steel material, and during simulation it was grounded at the potential of U = 0. The anode material of graphite was placed at the distance of 20 mm from tool bottom surface, and it was connected with the electric potential value of U = 60 V. The governing and boundary conditions used to perform the simulation study are given in Equations (1)–(3), respectively.
(1)$$\frac{\partial^{2}Ø}{\partial x^{2}}+\frac{\partial^{2}Ø}{\partial y^{2}}$$

Boundary conditions:Øc = 0(2)
Øa = U(3)
where, Øa and Øc are the values of electric potentials at the surface of auxiliary and tool electrode, respectively. U is the value of applied potential that was provided across the two electrodes.

During simulation, other electrical parameters including the conductivity of electrolyte, auxiliary electrode material, tool electrode material, and inter electrode distance were assumed constant throughout the simulation.

The extremely fine meshes were used to conduct the simulation, where the maximum and minimum sizes were of 40 μm and 0.08 μm, respectively. The mesh size was finalized after performing the repeatability and variation tests. The entire study was conducted on static mode conditions.

In order to analyze the performance of rough tools developed by RM-MT-EDM process, an experimental investigation was performed on ECSM system. The energy channelization index and specific energy were selected as response characteristics. The specific energy and energy channelization index (ECI) were calculated by using the formula given in Equations (4) and (5), respectively.
Specific energy = (V × I × tm × η)/((wi − wf)/tm)(4)
η = Ton/(Ton + Toff)
ECI = (LTT/DTT)/(LST/DST)(5)
where, V, I, tm, and η are the applied voltage, current, machining time and duty ratio, respectively. Ton and Toff are pulse on and off time durations, respectively. wi and wf are the weights of the work material before and after the machining.

A digital weighing machine (Shimadzu AUW220D, Kyoto, Japan) was used to measure the initial and final weight of work material. The L and D are the hole depth and diameter, respectively, wherein the subscripts ST and TT represents the smooth/conventional tool and rough tool electrode, respectively. A stereo zoom microscope (Make: Nikon SMZ745T, Nanyang, China) was used to measure the depth and diameter of the machined holes.

## 3. Results and Discussion

The results and discussion section of this manuscript comprise three phases. The first phase includes the fabrication of rough tools by RM-MT-EDM process. The second and third phases are dedicated to analyzing the performance of rough tools during the ECSM process.

### 3.1. Fabrication of Rough Tools

In this investigation, the thermal energy produced by RM-MT-EDM process is utilized to fabricate the rough surface over the tool electrodes. The mechanism of the tool texturing is illustrated in Figure 4. As mentioned earlier, in the tool roughening system, the copper wire brush and the SS-304 tool electrode are connected to the negative and positive terminals of the DC pulsed power supply. Thereby, in such an arrangement, the wires of copper brush and the tool electrode of SS-304 develop multiple circuits as shown in Figure 4. Here, each copper wire acts as a cathode and the opposite surface of the tool electrode act as an anode.

When the electric potential is applied across these multiple circuits, the high electric field is developed across the copper wire brush and the SS-304 tool electrode. Further, the electrons break loose from the surface of each copper wire. Due to the existence of a high electric field between the multiple circuits, the electrons start impelling from the wire electrodes toward the SS-304 tool electrode. The entry of electrons into the inter electrode space leads to the collision between electrons and the neutral molecules of the dielectric medium, and further, this collision results into ionization of dielectric medium. Due to ionization, a narrow channel of continuous conduction between each wire electrode and SS-304 tool electrode surface is developed (as depicted in Figure 4). Subsequently, these multiple conduction channels allow a momentary current impulse across the copper wire and the tool electrode. Eventually, the passages of momentary current impulses through these multiple conduction channels results in multiple sparks from each copper wire. The thermal energy liberated by these multiple sparks produce the micro cavities over the tool electrode surface. The morphology of the fabricated rough tool is depicted in Figure 5. In this figure, it can be seen that the rough tools had dimples (cavities) and humps which may be created due to the energy liberated by the sparks in random manners as reported by Jithin et al. [22].

During experimentation, the value of current was varied from 1 A to 4 A, while the other parameters were kept constant (Lift = 2; pulse on:pulse off = 170:67 μs; gap voltage = 60 V; tool rotation rate = 500 rpm). The constant value of process parameters was selected based on the pilot experiments, where a current was founded as one of the most significant parameters to effect the surface roughness value [11]. In order to get the repeatable conditions, each experiment was performed three times at individual parametric settings, the topo graphs of fabricated rough (rough) tools are depicted in Figure 6.

It can be seen from Figure 6 that with an increase in current value, the surface roughness of the tool electrode increases. The reason for this is the that increase in spark intensity causes an increase in the current value. Thereby, the energy penetrated by high intensity sparks resulting in the formation of deep craters over the tool electrode surface, and consequently it results into rough surfaces [23,24]. Here, TT 1 and TT 4 represent rough tools with a minimum and maximum surface roughness, respectively.

### 3.2. Electrical Characterization of Rough Tools

The results obtained from the simulations of smooth and rough surface tool electrodes are shown in Figure 7a–e. The effect of surface roughness on the electric field intensity is also presented in graphical form as depicted in Figure 7f. It can be seen from Figure 7f that the electric field intensity increases with an increase in surface roughness for the same value of applied voltage. The reason for this can be attributed to the fact that an increase of the electric field intensity causes an increase in the density of electrons at the sharp edges. Since sharp edges exhibits small space, a large number of charges thus become accumulated in the small space. It can also be observed that the electric field is more intensified at the edges of the tool electrode with a smooth surface (as depicted in Figure 7a). On the other hand, it can be seen from the Figure 7b–e that the distribution of electric field is more uniform and intensified from the bottom surface of the tool electrode with rough surface instead of the edges due to the uniform distribution of charges at the multiple edges. At the initial stage, the maximum electric field intensity was 3.1 × 105 V/m. However, at the final stage it increased up to 5.41 × 105 V/m. As compared to smooth surface tool electrode, the electric field intensity for rough tools is improved by 265.54%.

### 3.3. Energy Channelization Analysis of Rough Tools during ECSM

Further, to analyze the performance of rough tools developed by the RM-MT-EDM process, an experimental investigation was performed using ECSM facility. During experimentation, the applied voltage, pulse ratio, and electrolyte concentration were fixed at constant levels of 58 V, 2:2 ms and 20% respectively. Each experiment was performed thrice and the mean value of responses obtained from these three trials was considered as final quality characteristic. The effect of surface roughness of tool electrode on energy channelization index and specific energy is shown in Figure 8. The voltage signals captured during experimentation and their respective machined holes are also shown in Figure 9.

It was observed that the smooth tool (ST) electrode developed a thick gas film over the surface of tool electrode. Thereby, the breakdown of thick gas film results in low frequency and high intensity spark sparks as depicted in Figure 9. Further, the thermal energy liberated by these sparks evaporates the water from the electrolyte and leaves behind the molten salt in the machining zone. Thereby, the subsequent gas bubble formation starts from the side walls of tool electrode. The breakdown of gas film from tool side walls initiates side sparks and consequently channelizes the applied energy from the tool side walls instead of the bottom face of the tool electrode. Thus, the use of smooth tool electrodes increases the diameter of holes instead of the depth, and results in a low L/D ratio. It can be seen from the Figure 8 that the use of rough tools improves the energy channelization index and decreases the specific energy that is required to remove the 1 mg material in 1 s. As the surface roughness of tool electrode increases, the energy channelization index also increases. The reason thereof is the generation of high frequency sparks (as shown in Figure 9). The positioning of the rough tool over the work material with zero working gap creates the micro gaps between the upper face of the work material surface and the bottom face of the tool electrode as shown in Figure 10. Thus, the bubbles evolved from these micro gaps generate stable and small gas films due to their limited cavity size, and the subsequent gas film breakdown results in improved frequency of spark sparks. The decreased intensity of sparks with an increase in surface roughness as depicted in Figure 9 is evidence of the formation of small and stable gas film from the micro gaps of rough tools.

Beyond a surface roughness of 11.70 μm (refer tool electrode TT 3), with an increase in surface roughness the energy channelization index decreases and specific energy increases as shown in Figure 8. The main reason for this can be attributed to an increase in cavity size, which accompanies an increase in the roughness of the rough tool. The increased cavity size provides more space for the growth of hydrogen bubbles. Therefore, thick gas films develop on the bottom face of the rough tool. Further, the breakdown of these thick gas films generates low frequency, high intensity sparks (as shown in Figure 8). The energy liberated by these sparks evaporates water from the electrolyte and the molten salt leaves behind at the machining zone. Thus, the subsequent bubble growth starts from the side walls of rough tool instead of the bottom face. The film formation from tool side walls leads to side sparks, and consequently it enlarges the diameter of the hole instead of the depth. The impact of side discharges can also be seen from the micro hole images given in Figure 9. It can clearly be seen from the figure that the use of smooth tool results in micro hole fabrication with cracked edge surfaces. On the other hand, with the introduction of rough tools, the quality of micro holes improves and results into the formation of smooth edge holes. Apart from this, the inconsistency in material removal behavior can also analyzed from the deviations obtained in results. In case of smooth tools, the variation in results (energy channelization index and specific energy) is maximum and it reduces with the use of rough tools as depicted in Figure 8.

Further, on the basis of the experimental results, simulation observations, and spark characteristics obtained during experimentation, a process mechanism for ECSM has also been proposed with smooth and rough tool electrodes. The sequential progress in process mechanism of ECSM with smooth and rough tools is illustrated in Figure 10.

From the illustrated mechanism, it can be inferred that in case of smooth tool electrodes, the hydrogen gas bubbles evolve from the side walls of tool electrodes instead of the tool bottom due to the zero working gap enforced between the smooth tool bottom surface and work substrate. Whereas in the case of rough tools, the hydrogen bubble’s growth starts from the bottom surface of the tool due to the presence of micro cavities between tool bottom surface and work material. Thereby, the film formation starts from the bottom surface. In case of smooth tools, formation is initiated from the tool’s side walls. Subsequently, due to the existence of high electric field intensity at the edges of the smooth tool electrode, the spark initiates from the periphery of the tool electrode, and thereby the applied energy is channelized from the edges toward the center of the tool electrode. However, in case of rough tools, the electric field intensity is uniformly distributed over the entire bottom surface of the tool, and the subsequent breakdown of the gas film from the bottom surface initiates the spark discharging from the bottom surface of the tool electrode. Thus, it results in uniform energy channelization from the entire bottom surface of the tool electrode and, consequently a material removal takes place from the work surface. In the case of rough tools, due to the uniform material removal action from the tool bottom surface, the tool electrode penetrates into the work material easily for successive machining steps. In the case of smooth tools, tool electrodes could not proceed further till the material removal will not take place from the entire bottom surface. Thus, for successive machining steps, the applied energy is consumed in order to increase the diameter of the hole instead of its depth.

#### Future Scope

In the current investigation, the effect of tool roughness was only limited up to the discharge regime. However, in future, this study can further be extended for hydrodynamic regime. The effect of discharge energy on the heat affected zone and change in the chemical composition of work substate can also be investigated.

## 4. Conclusions

The present work focusses on the fabrication and performance analysis of rough tools. In this study, rough tools were fabricated by rotary mode multi tip electric discharge machining (RM-MT-EDM) process and, further, an electric field analysis was carried out by multi-physic COMSOL simulation software. These rough tools were utilized to machine micro holes by ECSM process. The mechanisms for surface roughening and ECSM process with smooth and rough tools have also been illustrated with appropriate schematics. The generalized conclusions drawn from the experimental investigations are given below:
The use of rough tools in ECSM process generates stable and thin gas film. Thereby, the breakdown of these stable and thin gas films results into the generation of high frequency spark discharges.The use of rough tools generates uniform electric field intensity over the surface of tool electrode and also improves the electric field intensity by 265.54% as compared to smooth surface tool electrodes.The tool electrodes with rough surface improves the energy channelization index by 248.40% than the smooth tools and also reduces the specific energy utilize to remove the per unit material in per unit time by 143.263%.The increase in surface roughness of rough tools increases the energy channelization index and decreases the consumption of energy required to remove the per unit material in per unit time. However, beyond a certain value (11.70 μm), the effect of increased surface roughness exhibits reverse trends due to an increase in the cavity size of the tool electrode.

## Figures and Tables

**Figure 1 materials-15-05598-f001:**
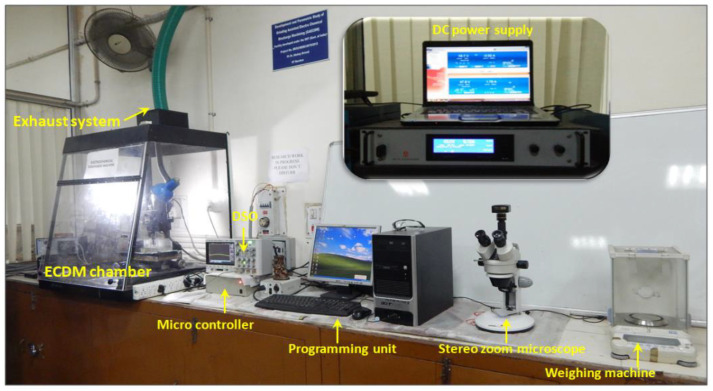
Pictorial view of ECSM machine.

**Figure 2 materials-15-05598-f002:**
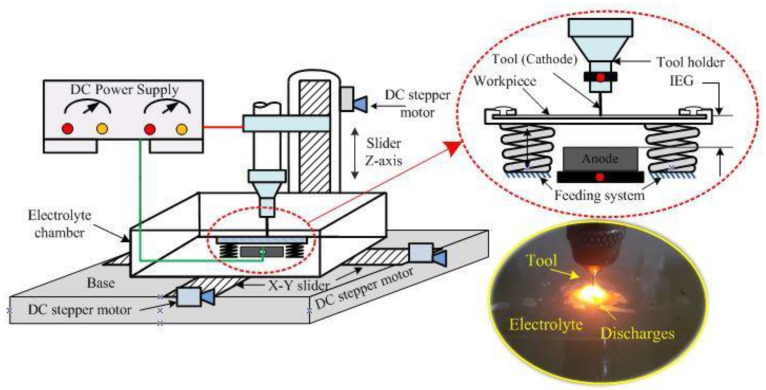
Schematic view of ECSM machine.

**Figure 3 materials-15-05598-f003:**
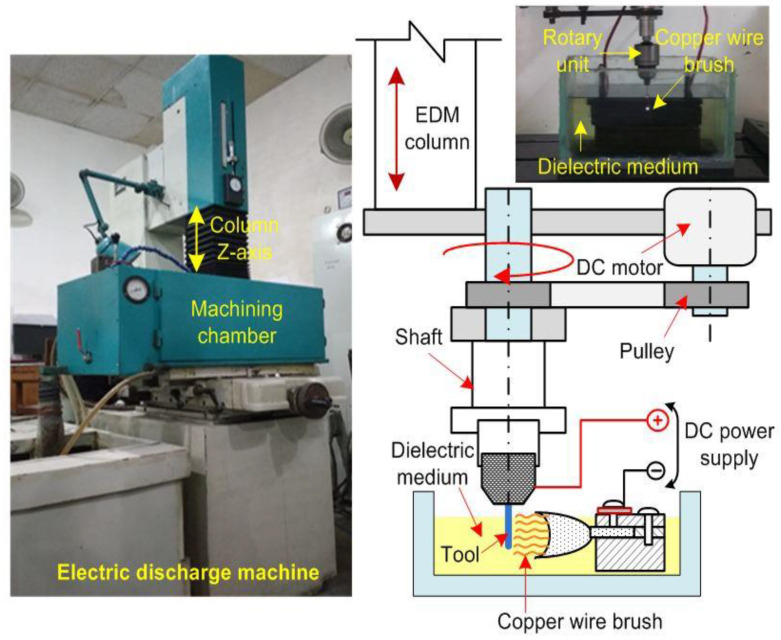
Pictorial view of the EDM machine and schematic view of the tool roughing system.

**Figure 4 materials-15-05598-f004:**
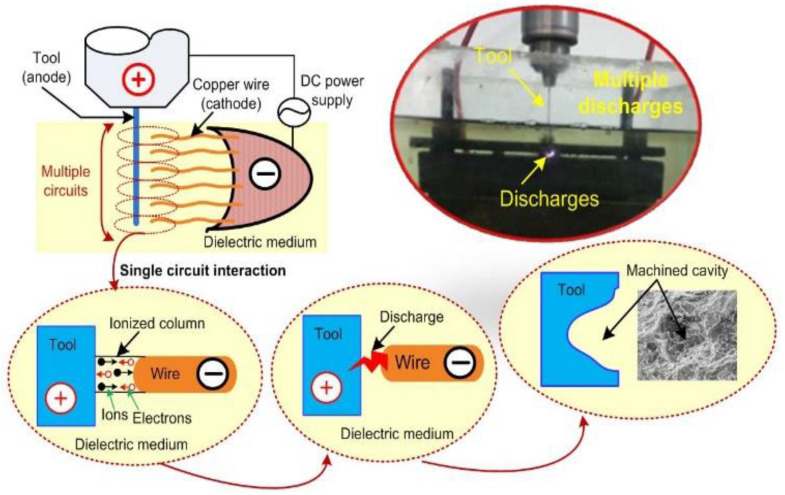
Mechanism for fabrication of rough tools by RM-MT-EDM process.

**Figure 5 materials-15-05598-f005:**
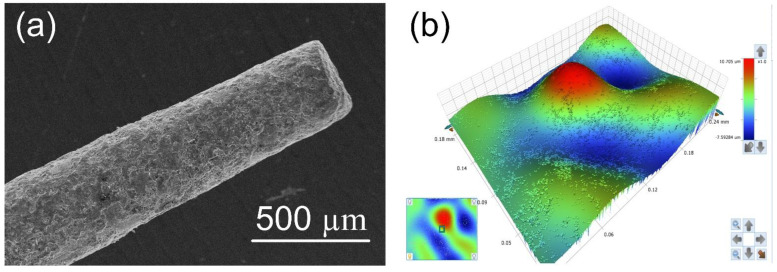
SEM image of (**a**) rough tool and (**b**) topographic view rough tool surface.

**Figure 6 materials-15-05598-f006:**
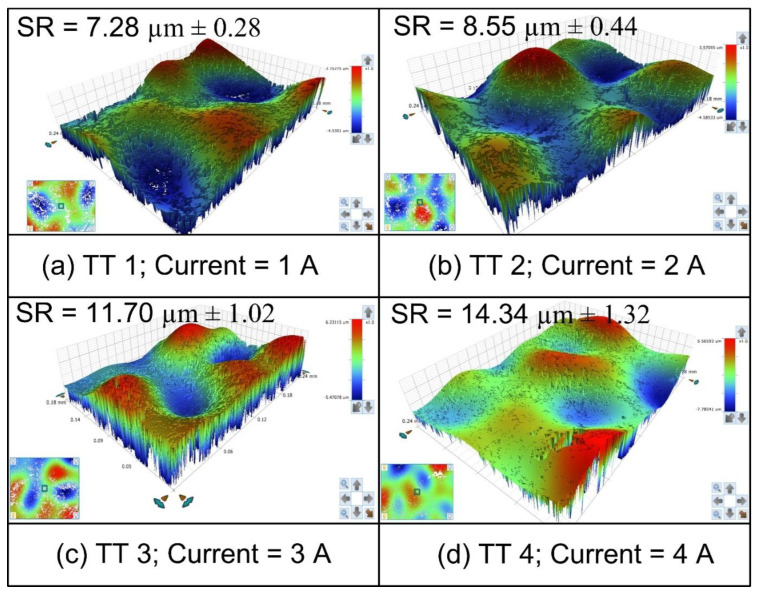
Effect of applied current values on the surface roughness of tools (**a**) current = 1A, (**b**) current = 2A, (**c**) current = 3A, and (**d**) current = 4A.

**Figure 7 materials-15-05598-f007:**
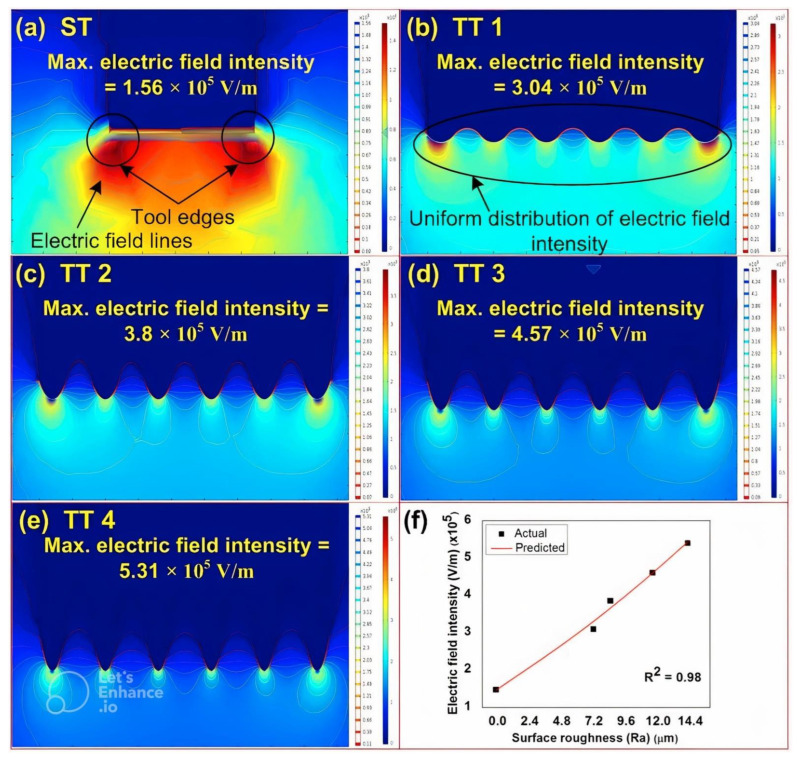
Effect of topography of tool electrode on electric field intensity (**a**) smooth tool, (**b**–**e**) tool with different roughness values, and (**f**) effect of surface roughness on electric field intensity.

**Figure 8 materials-15-05598-f008:**
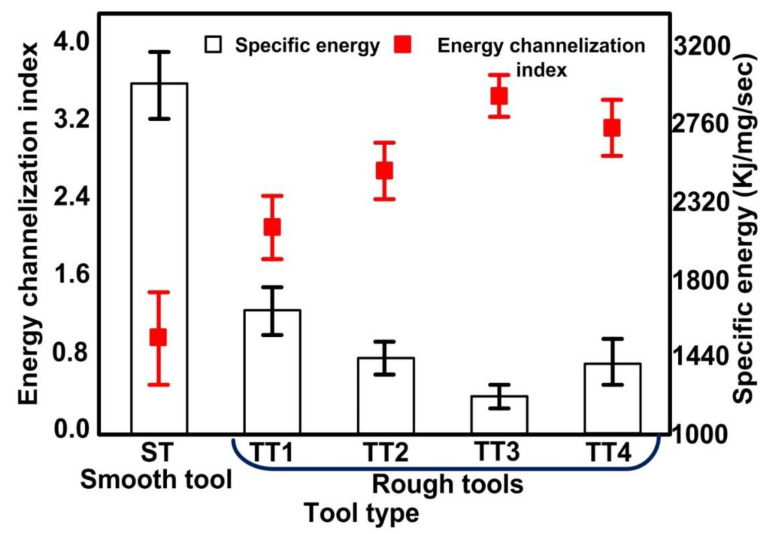
Effect of tool electrode topography on energy channelization index and specific energy.

**Figure 9 materials-15-05598-f009:**
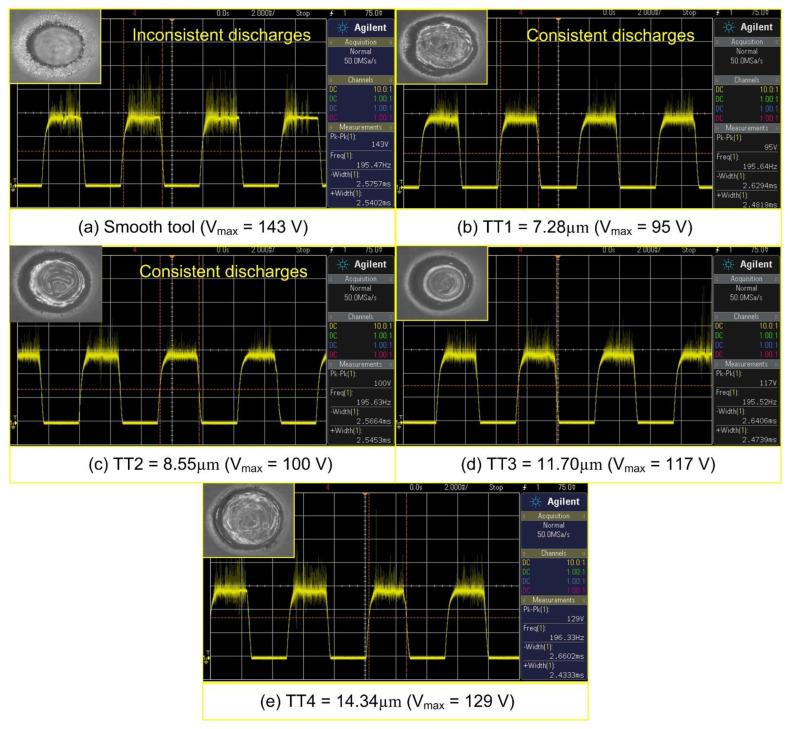
Effect of tool surface roughness on spark characteristics (Inset: machined micro holes).

**Figure 10 materials-15-05598-f010:**
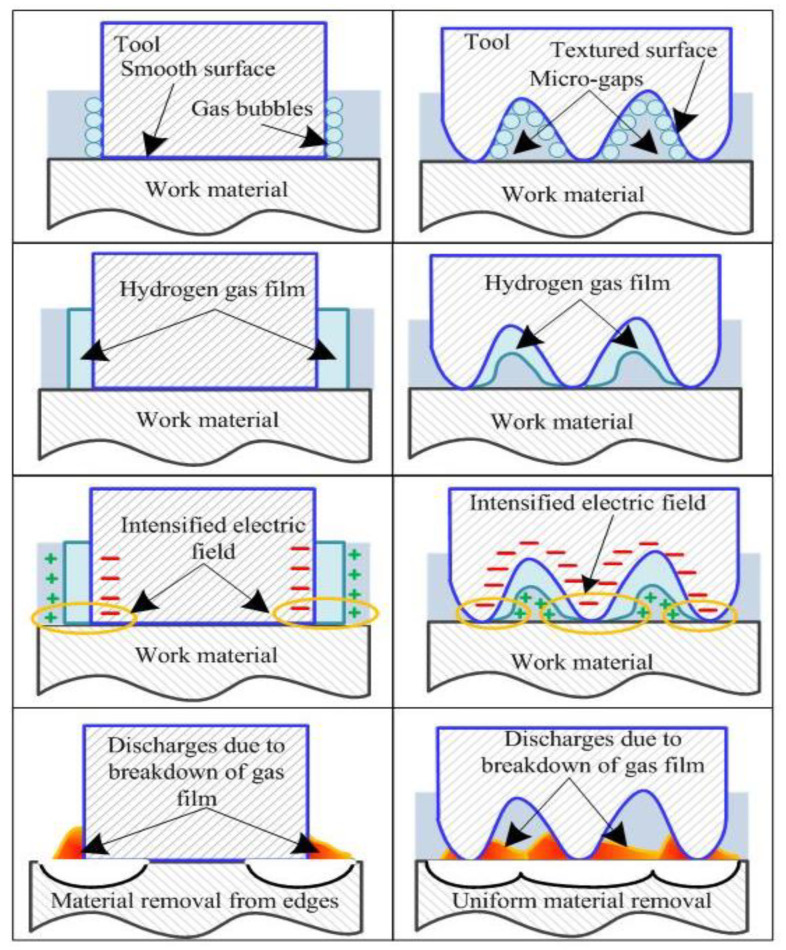
Mechanism of material removal with smooth and rough tool electrodes during ECSM process.

## Data Availability

Not applicable.

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
