# Peer review of "Energy Channelization Analysis of Rough Tools Developed by RM-MT-EDM Process during ECSM of Glass Substrates"

_materials, 2022, doi:10.3390/ma15165598_

Round 1

Reviewer 1 Report

1) The abstract and conclusion sections need to be improved.

2) The literature review is too general and some relevant papers dealing with the investigation on energy channelization analysis of rough tools developed by RM-MT-EDM process are missing.

3) In a current form, a novelty of the conducted research is unclear.

4) In present article, it needs to add the main research contributions of the present research compared with the existing research.

5) It is necessary to clearly indicate how to measure surface roughness.

6) In this article it is necessary to clearly point out what software is used for the simulation of electrostatic field intensity?

7) Additionally, it is necessary to add a section to explain in detail how to develop the electrostatic field simulation model.

8) The author wrote that “It can be seen from the Figure 7.f that the electric field intensity increases with an increase in surface roughness.”

It is necessary to provide the reasonable and sufficient explanations for the above research findings.

9) In Fig. 8, it is necessary to deeply and detailed analyze the effect of tool electrode topography on energy channelization index, specific energy and spark characteristics, and provide the more reasonable and sufficient explanation for the research results.

10) Similarly, it is necessary to provide the deeper discussion for the mechanism of material removal with smooth and rough tool electrodes during ECSM 268 process.

11) It is found that the manuscript is incomplete in all sense and the main discussion section is minimum.

12) The results are mainly presented by figures. It is more important to provide the sufficient and reasonable explanation for all the research results. This is also the weakest aspect of the study.

13) Results and discussion should be modified accordingly to the aim of the article.

14) The limitations of the study are not considered.

      15) The format of all the references should be modified according to the journal guidelines.

Author Response

Authors are very grateful to the reviewer’s for his/her valuable suggestions towards the improvement of manuscript quality. All the suggestions have been incorporated in revised version of the manuscript, and the changes have also been highlighted.

Reviewer 2 Report

The paper is an interesting dissertation on the energy analysis of tools used in ECSM of glass substrate

The paper is well organised, just an advice regarding Fig. 5. The magnification of the micrograph seems to be too low to analyse the actual condition of the surface. Please change it or explain in the text how you determine dimples and humps.

Author Response

(The authors gave the same response as above.)

Reviewer 3 Report

The results are good but deep discussion is needed. 

Indeed, there is 9 of 15 references for the same author (you). I think more references related to this work should be added  and the findings of this work should be discussed and linked  with other findings and methods. 

Author Response

(The authors gave the same response as above.)

Reviewer 4 Report

The submitted article titled "Energy channelization analysis of rough tools developed by  RM-MT-EDM process during ECSM of glass substrates" is interesting. However, following concerns may need to be addressed for that manuscript to be accepted:

1. In introduction section authors should elaborate the working of EDM, its significance and rationale for its selection. That will be helpful for the reader to build the context. Following article may be used for adding explanation of EDM:

a. An Optimalization Study on the Surface Texture and Machining Parameters of 60CrMoV18-5 Steel by EDM

b. Analyzing micromachining errors in EDM of Inconel 600 using various biodegradable dielectrics

c. EDM of Ti6Al4V under nano-graphene mixed dielectric: a detailed roughness analysis

d. Investigating cryogenically treated electrodes’ performance under modified dielectric (s) for EDM of Inconel (617)

e. EDM of Ti-6Al-4V under nano-graphene mixed dielectric: a detailed investigation on axial and radial dimensional overcuts

f. Toward the Targeted Material Removal with Optimized Surface Finish During EDM for the Repair Applications in Dies and Molds

2. In materials and method section authors need to add information regarding the rotary mechanism and its working parameters as they have vital importance as far as output of EDM is concerned. The justification regarding the selection of those parameters should also be cited.

3. Authors should highlight the number of replicates performed for completing the experimentation. Also standard deviation needs to be added as well.

4. A comparison with the results achieved during the ECSM needs to be compared with the existing literature as well.

5. A more detailed reasoning required to be mentioned in the result and discussion section.

6. Conclusion section requires refinement.  

Author Response

(The authors gave the same response as above.)

Round 2

Reviewer 1 Report

The authors have considered some comments of reviewers. Nevertheless, some aspects of the comments were ignored.

1) It is necessary to add more experimental details related to how to measure surface roughness.

2) It is necessary to add more simulation details related to how to develop the electrostatic field simulation model.

3) In Fig. 10, it is necessary to provide the deeper discussion for the mechanism of material removal with smooth and rough tool electrodes during ECSM process.

Author Response

(The authors gave the same response as above.)

Reviewer 3 Report

The authors responded now to most required suplementary information, the article can be published now in its revised form

Author Response

(The authors gave the same response as above.)

Reviewer 4 Report

Authors have incorporated  the suggestions

Author Response

(The authors gave the same response as above.)
